# Recent Data about the Use of Corticosteroids in Sepsis—Review of Recent Literature

**DOI:** 10.3390/biomedicines12050984

**Published:** 2024-04-30

**Authors:** Alexandra Lazar

**Affiliations:** Anesthesiology and Intensive Care Department, “George Emil Palade” University of Medicine, Pharmacy, Science and Technology from Tirgu Mures, 540142 Targu Mures, Romania; alexandra.lazar@umfst.ro

**Keywords:** corticosteroids, sepsis, septic shock, critically ill, intensive care

## Abstract

Sepsis, characterized by life-threatening organ dysfunction due to a maladaptive host response to infection, and its more severe form, septic shock, pose significant global health challenges. The incidence of these conditions is increasing, highlighting the need for effective treatment strategies. This review explores the complex pathophysiology of sepsis, emphasizing the role of the endothelium and the therapeutic potential of corticosteroids. The endothelial glycocalyx, critical in maintaining vascular integrity, is compromised in sepsis, leading to increased vascular permeability and organ dysfunction. Corticosteroids have been used for over fifty years to treat severe infections, despite ongoing debate about their efficacy. Their immunosuppressive effects and the risk of exacerbating infections are significant concerns. The rationale for corticosteroid use in sepsis is based on their ability to modulate the immune response, promote cardiovascular stability, and potentially facilitate organ restoration. However, the evidence is mixed, with some studies suggesting benefits in terms of microcirculation and shock reversal, while others report no significant impact on mortality or organ dysfunction. The Surviving Sepsis Campaign provides cautious recommendations for their use. Emerging research highlights the importance of genomic and transcriptomic analyses in identifying patient subgroups that may benefit from corticosteroid therapy, suggesting a move toward personalized medicine in sepsis management. Despite potential benefits, the use of corticosteroids in sepsis requires careful consideration of individual patient risk profiles, and further research is needed to optimize their use and integrate genomic insights into clinical practice. This review underscores the complexity of sepsis treatment and the ongoing need for evidence-based approaches to improve patient outcomes.

## 1. Introduction

Sepsis is recognized as a critical condition characterized by life-threatening organ dysfunction resulting from a maladaptive host response to infection. Septic shock is a more severe form of sepsis, distinguished by profound circulatory, cellular, and metabolic dysfunctions that significantly elevate mortality risk beyond that of sepsis alone [1]. These conditions are part of a growing global health crisis, presenting complex challenges for emergency physicians due to their rising incidence and the intricate interplay of pathophysiological, molecular, genetic, and clinical factors [2]. Since the initial consensus definition in 1991 (Sepsis-1), the global occurrence of sepsis and septic shock has been on an upward trajectory, with estimates indicating around 49 million sepsis cases and 11 million sepsis-related fatalities worldwide in 2017 [3].

Currently, the development of sepsis is understood to involve a complex interplay of numerous pro-inflammatory and anti-inflammatory mediators. Recent advances have also shed light on cellular changes specific to sepsis, with a growing emphasis on the role of microcirculation in the transition from sepsis to septic shock. In this framework, the endothelium emerges as a pivotal element in sepsis pathophysiology due to its critical functions in controlling microcirculation and regulating coagulation, as well as inflammatory and anti-inflammatory pathways [4,5]. The endothelial glycocalyx, made up of proteoglycans and glycoproteins, plays multiple roles, including acting as a physical barrier to control vascular permeability, facilitating leukocyte and platelet adhesion, and moderating inflammatory and anti-inflammatory responses. However, the structural integrity of the glycocalyx can be compromised—referred to as “glycocalyx shedding”—through exposure to oxidants, cytokines, exotoxins, and endotoxins from bacteria. This degradation leads to the migration of leukocytes across the endothelium and heightened vascular permeability, resulting in edema that increases interstitial pressure and exacerbates the impairment of tissue perfusion [6].

## 2. Corticosteroids as a Therapeutic Option

Over fifty years since the inaugural randomized, controlled trial examining use of corticosteroids for severe infections, their widespread use among physicians continues despite significant disagreements among experts regarding their risk-to-benefit ratio [7]. The widespread and enduring use of corticosteroids in treating severe infections can be attributed to the immediate and observable improvement in critical conditions like shock and respiratory failure observed in clinical settings. The ongoing debate among experts over the use of corticosteroids is not driven by new scientific evidence but by differing interpretations of existing data.

Corticosteroids are recognized for their extensive immunosuppressive properties, which can elevate the risk of infectious complications for patients. Moreover, stress often triggers an immediate surge in inflammatory mediators, leading to a systemic inflammatory response and significant suppression of the immune system [8].

### The Rationale for Using Corticosteroids

-Overactivity of proinflammatory pathways relative to endogenous glucocorticoid activity

Experts widely agree that uncontrolled systemic inflammation is a fundamental aspect of severe sepsis, significantly influencing the advancement of organ failure and death. The management of inflammation involves a complex interaction between the neuroendocrine and immune systems [9]. On a microscopic scale, equilibrium within the body is maintained through the interplay of two dynamic systems. The nuclear factor kappa B (NF-κB) pathway triggers the release of substances that promote inflammation, while the synergy between glucocorticoids and the glucocorticoid receptor alpha (G-GRα) complex serves to suppress inflammatory reactions [10]. In all cells, these processes are typically dormant; the NF-κB pathway is restrained by its inhibitor, inhibitory factor kappa B (I-κB), and regulation of the G-GRα complex is achieved by maintaining an equilibrium between its alpha and beta forms. When these systems successfully counteract each other, stability is preserved. Nonetheless, an inclination toward the activation of NF-κB may result in uncontrolled inflammation [10,11]. 

In cases of extended acute respiratory distress syndrome or septic shock in patients, a disproportion has been observed, with NF-κB being excessively active about the G-GRα complex. This imbalance is recognized as a contributing element to the damage of cells, tissues, and organs. The root reasons for adrenal insufficiency during critical illnesses have been extensively investigated in separate research [12]. Influences including drugs that alter cortisol metabolism, harm to the hypothalamic–pituitary–adrenal axis, and stimulation of the inducible nitric oxide synthase (iNOS) enzyme through cytokines contribute to the death of neuroendocrine cells. This sequence of events subsequently leads to a decreased responsiveness to ACTH (adrenocorticotropic hormone) and glucocorticoids. It has been noted that people who overcome critical illnesses typically regain normal endocrine function within a few weeks to months following their hospital discharge [12].

-Molecular mechanisms through which glucocorticoids act align seamlessly with the underlying pathological processes of sepsis

Glucocorticoids function via both genomic and non-genomic pathways. The non-genomic effects, manifesting within minutes after exposure to glucocorticoids, include decreased platelet aggregation and cell adhesion, reduced activity of intracellular phosphotyrosine kinases, and increased expression of annexin 1 on the cell surface [13]. The rapid effects are believed to result from glucocorticoids interacting with locations on the cell membrane. Conversely, the genomic activities of glucocorticoids are generally divided into two categories: transrepression and transactivation effects [14]. The prevailing view is that glucocorticoids primarily modulate rather than simply suppress immune cell function [15]. 

Microarray studies have shown that exposure to glucocorticoids activates a greater number of genes than it represses, highlighting their intricate impact on gene expression. Further research has also demonstrated that glucocorticoids encourage the development of specific anti-inflammatory monocyte subtypes, which rapidly migrate to areas of inflammation. Moreover, glucocorticoids extend the survival of these monocytes by activating anti-apoptotic pathways through the A3 adenosine receptor. This detailed function of glucocorticoids in regulating immune responses proves particularly advantageous in controlling the excessive inflammation seen in sepsis [16,17]. 

-Cardiovascular stability in sepsis

Corticosteroids facilitate the retention of sodium by affecting both mineralocorticoid and glucocorticoid receptors, a process that helps to combat the hypovolemia observed in the early phases of sepsis. Moreover, corticosteroids contribute to the retention of sodium and water within the blood vessel walls, which increases systemic vascular resistance. Through non-genomic actions, corticosteroids can quickly boost the responsiveness of blood vessels to alpha agonists in a matter of minutes to hours, leading to elevated mean arterial pressure and systemic vascular resistance [18]. Corticosteroids also contribute to blocking ATP-dependent potassium channels. Their ability to dampen the gene expression for iNOS and cyclooxygenase II helps to maintain an enhanced response to catecholamines for an extended period, spanning several days. The successful reinstatement of vascular responsiveness to vasopressors might be intricately linked to the degree of disparity in the activities of NF-κB and the G-GRα complex [19]. Administering moderate doses of hydrocortisone in septic shock patients led to improved capillary density and blood flow, effects that were observed as soon as one hour after treatment [20]. The improvement in microcirculation is believed to stem from heightened activity of the endothelial form of nitric oxide synthase, facilitated by a mechanism involving the mitogen-activated protein kinase/Akt pathway [21]. Moreover, corticosteroids might play a role in restoring the inherent fluctuations present in the cardiovascular system [22].

-Organ restoration and ICU length of stay

In septic shock patients, glucocorticoids were found to suppress the release of tumor necrosis factors from vascular and smooth muscle tissues [23]. Additionally, by the fifth day of administering hydrocortisone, there was a total inhibition of NF-κB activity in peripheral mononuclear cells. Corticosteroids have proven successful in diminishing renal iNOS activity after endotoxemia, aiding in the prevention of hypoxic injury to the kidney cortex, improving oxygen supply to the kidneys, and ultimately normalizing the use of oxygen by the kidneys [24]. Similarly, corticosteroids have been shown to enhance the permeability of the glomerular endothelium in septic shock patients [25]. Corticosteroids have a beneficial effect on the cardiovascular, pulmonary, hepatic, and renal systems. Consequently, patients undergoing corticosteroid treatment may rapidly transition off vasopressor support and mechanical ventilation, facilitating their earlier release from the intensive care unit [26].

Although so much evidence about the good side of corticosteroids is to be found in the literature, their administration in sepsis or septic shock is not yet convincing. The latest guidelines from the Surviving Sepsis Campaign for the use of corticosteroids in adults with septic shock recommend intravenous hydrocortisone at a daily dosage of 200 mg. This dosage is typically delivered as 50 mg, given intravenously every 6 h or through a continuous infusion. It is recommended to start this corticosteroid treatment at least 4 h after the initiation of norepinephrine or epinephrine, at a dose of ≥0.25 mcg/kg/min in adults. This is categorized as a weak recommendation with a moderate quality of evidence [2].

This review’s objective is to present the current evidence, based on thoroughly conducted and extensive studies, on the use of corticosteroids in sepsis and septic shock with the scope of creating a clearer image of their use in septic and septic shock patients.

## 3. Materials and Method

For the study research, the PubMed database was searched using the MesH terms “sepsis”, septic shock”, and “corticosteroids”. The search retrieved 26,577 results, which, after applying filters such as being published in the last five years, randomized controlled studies, exclusion of books, abstracts, and no free full text available, written in a language other than English, as well as reviews and meta-analyses, were narrowed to 12 multicentric, randomized control studies. The flow chart below presents the study’s search process (Figure 1).

The data processed from the final included articles were as follows: if the study was multicentric, the number of patients included, the intervention, the main results, and the conclusions of the study.

## 4. Results

Our findings are presented in Table 1.

The dosages of the corticosteroid treatments and their duration from the selected studies are presented below in Table 2.

## 5. Discussion

While corticosteroids are considered for use in certain cases of sepsis and septic shock, their administration is treated with caution due to the possibility of adverse effects, increased risk of exacerbating infections, and unpredictable reactions among individual patients. The decision to employ corticosteroids in treating sepsis requires a nuanced assessment of the potential advantages and dangers for each patient. Corticosteroids, which are potent anti-inflammatory agents, can adjust the immune system’s response. Given the severity of sepsis—a critical, life-endangering reaction to infection that may cause tissue harm, organ failure, and mortality—the application of corticosteroids has sparked significant discussion and investigation.

Some of the reasons for which corticosteroids are included in the last line of sepsis, aspects which constitute the disadvantages of their use in septic shock, include:-Corticosteroids can suppress the immune system. In the setting of sepsis, where the body is fighting a severe infection, further suppression of the immune response may be counterproductive, potentially allowing the underlying infection to worsen [41].-Due to their immunosuppressive effects, corticosteroids can increase the risk of secondary infections, which can complicate or exacerbate the patient’s condition [42].-Corticosteroids can increase blood sugar levels, which could complicate the management of septic patients, especially those with diabetes or those who develop stress-induced hyperglycemia because of their critical illness. One detail is very important about this statement, namely that most of these glucose level impairments related to corticosteroids in septic patients appear mainly when boluses are administered [43].-They can also cause fluid retention and electrolyte imbalance, potentially exacerbating sepsis-induced organ dysfunction, such as acute kidney injury [44].

## 6. Different Combinations of Corticosteroids

Despite the theoretical benefits of corticosteroids, vitamin C, and thiamine combination therapy, including anti-inflammatory and antioxidant properties and the potential to preserve endothelial function, the trials found no significant difference in primary outcomes like the change in the SOFA score at 72 h or secondary outcomes such as kidney failure, 30-day mortality, ventilator-free days, and ICU-free days between the intervention and placebo groups. However, the intervention group did have a statistically significant increase in the number of shock-free days compared to the placebo group [31,35,36].

These trials highlight the ongoing debate regarding the utility of ascorbic acid, hydrocortisone, and thiamine in septic shock treatment. Despite initial enthusiasm based on smaller, single-center studies, larger multicenter trials have not demonstrated a significant benefit in reducing organ failure, mortality, or other critical outcomes with this triple therapy therapy. The only consistent finding across trials has been a reduction in the duration of vasopressor support, suggesting a potential role for triple therapy in specific patient subgroups or phenotypes, rather than a universal treatment for all septic shock patients [27,28,29]. 

The study by Marik et al. published in *Chest* in June 2017 investigated the effects of a combination treatment involving hydrocortisone, vitamin C, and thiamine on patients with severe sepsis and septic shock. This retrospective before–after study involved comparing the outcomes of 47 consecutive septic patients treated with this combination therapy against 47 historical control patients who received standard of care.

The main findings of the study were significant. The treatment group that received the combination therapy showed a remarkable reduction in mortality compared to the control group (8.5% vs. 40.4%). Furthermore, the patients treated with the combination therapy also demonstrated a reduction in the duration of vasopressor administration, indicating an improvement in vascular responsiveness.

The authors concluded that the administration of hydrocortisone, vitamin C, and thiamine appears to be safe and could lead to a significant reduction in mortality in patients with severe sepsis and septic shock. However, they also suggested that these findings should be validated by randomized controlled trials to confirm the efficacy and safety of this treatment approach [45].

These results underscore the complexity of treating septic shock and the importance of ongoing research to refine and discover effective therapies. Current evidence does not support the widespread use of triple therapy to improve mortality or organ dysfunction in septic shock [27,28,29,31,35,36]. Future trials may provide further insight into whether there are specific patient populations who could benefit from this therapy.

## 7. Genomics in Sepsis Treatment

An emerging source of significant results in corticoid use in sepsis appears to be genomic subtyping and treatment personalization [33,39]. Wong HR et al. focused on a crucial component of personalized medicine in treating sepsis, especially in cases of septic shock. Their study examined the impact of corticosteroids on patients identified with a particular endotype of septic shock, known as endotype A,. This focus emphasizes the growing recognition and significance of genomics in the treatment of sepsis, showcasing how personalized healthcare strategies are becoming increasingly important in managing this complex condition [33].

The study by Wong and colleagues marks an important advancement in the treatment of sepsis, offering critical perspectives on the utilization of genomic data to enhance patient results. It underscores the shift toward a personalized treatment paradigm in sepsis management, emphasizing the importance of considering an individual’s genetic profile to refine therapeutic approaches. This investigation illuminates both the promising advantages of personalized medicine and the obstacles and factors that need attention as genomics increasingly influences clinical decisions in sepsis and beyond [33].

The research presented by Antcliffe and colleagues represents a significant step forward in the application of genomic technologies in sepsis treatment. By demonstrating how transcriptomic profiles can differentiate responses to corticosteroids among sepsis patients, this work paves the way for more personalized, effective, and safe treatment strategies [39]. This research utilized data from the VANISH (Vasopressin vs. Norepinephrine as Initial Therapy in Septic Shock) randomized trial to examine how genomic technologies, especially transcriptomics, might redefine sepsis management, with a focus on corticosteroid administration [40]. The research uncovered distinct transcriptomic patterns in sepsis patients, linking these patterns to variations in how individuals respond to corticosteroid treatments. These findings provide valuable insight into the biological mechanisms and pathways that are activated in sepsis and their interaction with treatment methods.

The study highlights the capacity of transcriptomics to personalize corticosteroid therapy for sepsis patients. Identifying patients likely to respond favorably to corticosteroids through their transcriptomic profile allows for more precise medical decisions, potentially enhancing patient outcomes and minimizing negative side effects [39].

Transcriptomic analyses shed light on the specific ways in which corticosteroids affect patients with septic shock, offering a clearer understanding of the drugs’ biological actions. Such insight could guide the discovery of biomarkers for treatment prediction and the development of novel treatment avenues.

## 8. Conclusions

Recent advancements underscore a move toward individualized medicine in managing sepsis, highlighting the potential of genomic and transcriptomic analyses to inform more customized treatment approaches. Such insight could significantly enhance patient outcomes by identifying those who are likely to benefit from corticosteroid therapy and those who may face detrimental effects. However, integrating genomic personalization into clinical practice faces considerable challenges, including the need for rapid, accurate genomic assessments and the creation of clinical protocols to effectively use genomic information.

There is a pressing call for additional studies to validate and expand the use of genomic and transcriptomic data in guiding therapeutic decisions for sepsis and septic shock. This includes exploring further genomic indicators and omics techniques to improve the customization of care and patient outcomes.

In essence, while corticosteroids are crucial in treating sepsis and septic shock, their use requires careful consideration of potential benefits against risks. The introduction of genomic technologies offers the promise of treatment plans tailored to individual patients, yet fully realizing this potential necessitates ongoing research and overcoming current barriers to implementation.

## Figures and Tables

**Figure 1 biomedicines-12-00984-f001:**
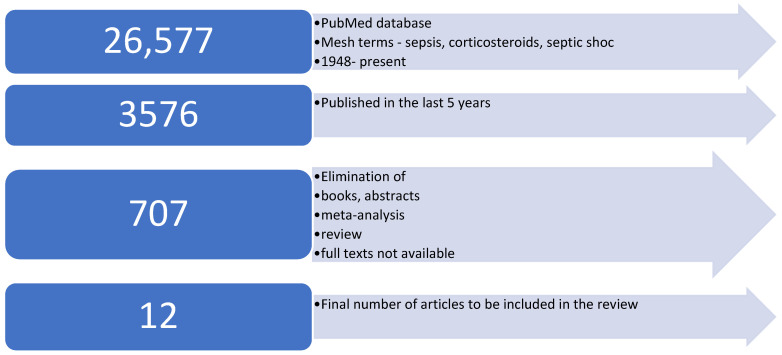
Flow chart of the study’s search process.

**Table 1 biomedicines-12-00984-t001:** The findings of this study.

Study/Year Published	Multicentric Yes/No	No. of PTS	Intervention	Results	Conclusion
Resuscitation With Vitamin C, Hydrocortisone, and Thiamin in Children With Septic Shock: A Multicenter Randomized Pilot Study [27] 2024	YES	60	Patients assigned to the group receiving vitamin C, hydrocortisone, and thiamin were administered 30 mg/kg per dose of vitamin C (up to a maximum of 1500 mg per dose as sodium ascorbate, sourced from biological therapies) intravenously every 6 h, 1 mg/kg per dose of hydrocortisone (with a maximum of 50 mg per dose) intravenously every 6 h, and 4 mg/kg per dose of thiamin (up to a maximum of 200 mg per dose) intravenously every 12 h, starting immediately after they were randomized. The treatment was continued for 7 days or until any of the following occurred: the shock was resolved, the patient died, the patient was discharged from the pediatric intensive care unit (PICU), or if any major adverse events attributed to the treatment were observed.	By the 28th day, participants in the intervention group had a median of 20.0 organ dysfunction-free days, compared to 21.0 days in the group receiving standard care. The median duration without the need for inotrope support within the first 7 days was 6.3 days for those in the intervention group and 5.9 days for those receiving standard care. Within the first 28 days post randomization, 15% (4 out of 27) of patients in the intervention group passed away, as opposed to 6% (2 out of 33) in the standard care group. The median stay in the oediatric intensive care unit (PICU) was 5.3 days for the intervention group and 6.9 days for the standard care group.	A pragmatic trial on high-dose vitamin C, hydrocortisone, and thiamin in children with septic shock admitted to PICU is feasible.
Evaluation of hydrocortisone, vitamin C, and thiamine for the treatment of septic shock: a randomized controlled trial (The HYVITS Trial) [28] 2023	YES	106	The three-drug treatment involved administering 1.5 g of intravenous vitamin C every 6 h for 4 days or until the patient was discharged from the ICU, whichever came first, 50 mg of hydrocortisone every 6 h for 7 days or until ICU discharge, followed by a gradual reduction over 3 days, and 200 mg of intravenous thiamine every 12 h for 4 days or until ICU discharge, whichever occurred sooner. This was in comparison to providing only standard care to patients experiencing septic shock.	No significant statistical differences were observed between the group receiving triple therapy and the control group in terms of: Mortality within the hospital at 60 days. Rates of discharge from the hospital. The length of vasopressor use among those who survived. The amount of time spent on mechanical ventilation. The variation in SOFA (sequential organ failure assessment) score at 72 h. The requirement for renal replacement therapy was comparable between the two groups. Outcomes related to safety. However, a higher incidence of failure to wean off mechanical ventilation was noted in the triple therapy group (80% compared to 64.1%).	In patients with septic shock who needed vasopressor support, the combined use of hydrocortisone, vitamin C, and thiamine did not lead to a reduction in 60-day in-hospital mortality, nor did it shorten the length of vasopressor use or lower SOFA scores at 72 h.
Effects of hydrocortisone combined with vitamin C and vitamin B1 versus hydrocortisone alone on microcirculation in septic shock patients: A pilot study [29] 2023	NO (pilot study)	27	The treatment group received a combination of hydrocortisone, vitamin C, and vitamin B1, in addition to standard care. The control group was administered hydrocortisone as a standalone treatment, alongside standard care.	Between the treatment and control groups, no statistically significant differences were observed in: sPVD (skin perfusion video densitometry) at the outset. Baseline measures of sPPV (skin perfusion pressure variability), sTVD (skin total vessel density), or MFI (microcirculatory flow index). Time to peak (TTP) or mean transit time (MTT) 24 h post treatment in the treatment group versus the control group. However, after adjusting for baseline perfusion index (PI) and renal blood flow (RBF), both PI and RBF were significantly greater in the treatment group than in the control group 24 h after receiving treatment. Lactate levels were notably lower in the treatment group compared to the control group 24 h post treatment. No significant variance was found in the doses of norepinephrine administered at baseline, 4 h, and 24 h post treatment between the two groups. The treatment group exhibited significantly higher values for sPPV, sTVD, and MFI compared to the control group 24 h after treatment.	The combination therapy was significantly more effective at enhancing microcirculation in patients with septic shock than hydrocortisone alone.
Corticotropin-stimulated steroid profiles to predict shock development and mortality in sepsis: From the HYPRESS study [30] 2022	YES	206	Corticotropin test for all included patients.	In healthy participants, the response to corticotropin stimulation varied widely, showing a highly dynamic reaction across all analyzed steroids and their precursors. When comparing the sepsis group to healthy individuals, baseline corticosterone levels were similar, but cortisone levels were notably lower in the sepsis group. Following corticotropin stimulation, sepsis patients exhibited the same levels of the precursor 17-OH progesterone and significantly elevated levels of 11-desoxycorticosterone, 11-desoxycortisol, and cortisol compared to healthy subjects. However, the rise in corticosterone was significantly less in sepsis patients than in healthy subjects. Comparing healthy individuals with patients having severe sepsis revealed that 50% of the sepsis patients (those not experiencing shock) had a below normal increase in corticosterone. Analysis of outcomes, specifically in-hospital mortality within the placebo group, indicated that sepsis patients who did not survive had significantly smaller increases in corticosterone than those who did survive, pointing to a more severe disruption in mineralocorticoid metabolism among the non-survivors.	Steroid profiling indicated that the mineralocorticoid pathway was often more affected than the glucocorticoid pathway in the body’s stress response to sepsis. Patients exhibiting higher levels of glucocorticoids relative to mineralocorticoids following corticotropin stimulation had a higher likelihood of progressing to septic shock and succumbing in the hospital. The cortisol-to-corticosterone ratio post corticotropin stimulation can serve as a predictor for critical outcomes like the onset of shock and death in sepsis cases. This ratio may also guide the application of hydrocortisone treatment.
Early administration of hydrocortisone, vitamin C, and thiamine in adult patients with septic shock: a randomized controlled clinical trial [31]. 2022	NO	408	Patients suffering from septic shock were mainly managed with an aggressive fluid resuscitation strategy, appropriate antibiotics, and vasoactive medications, following the guidelines of the Surviving Sepsis Campaign. Norepinephrine was chosen as the initial vasoactive medication. The treatment group received a regimen of hydrocortisone (200 mg daily), vitamin C (2 g every 6 h), and thiamine (200 mg every 12 h) for 5 days or until patients were discharged from the ICU. Conversely, the placebo group was given an equivalent volume of 0.9% saline from a placebo container, adhering to the same administration schedule.	In the intention-to-treat (ITT) analysis, there was no significant difference in 90-day survival between the groups. In the per-protocol (PP) analysis, 90-day survival also showed no significant difference between the groups. Within the ITT population, there was no noticeable difference in 28-day mortality, or ICU mortality, between those receiving the intervention and those receiving a placebo. For the PP population, there was no significant variance in 28-day all-cause mortality, mortality upon ICU discharge, or hospital discharge between the intervention and placebo groups. The ITT analysis revealed that the rate of shock reversal was comparable between the intervention and placebo groups. There was no statistically significant difference observed in the 72 h delta SOFA score, nor in the 28-day cumulative days free from the ICU, vasopressors, or ventilatory support between the two groups. No notable differences were found regarding the length of stay (LOS) in the ICU or the hospital between the intervention and placebo groups. In the PP analysis, the proportion of patients experiencing shock reversal was alike in both the intervention and placebo groups. There was no statistical significance found in the 28-day cumulative ICU-free days between the groups. Regarding adverse events within the safety population, the most frequently occurring serious adverse events were severe hypernatremia and fluid overload. A disturbance in blood glucose levels was noted in 27 patients within the intervention group	In individuals with septic shock, the combination of hydrocortisone, vitamin C, and thiamine did not demonstrate a reduction in 90-day mortality when compared to a placebo. Based on these findings, the regular application of this therapy mix for adult patients experiencing septic shock is not advocated.
Effect of 12 mg vs. 6 mg of Dexamethasone on the Number of Days Alive Without Life Support in Adults With COVID-19 and Severe Hypoxemia [32] 2021	YES	1000	Two study groups: Received 6 mg of dexamethasone. Received 12 mg of dexamethasone.	Participants in the 12 mg dexamethasone group experienced a median of 22.0 days alive without the need for life support, compared to 20.5 days in the 6 mg dexamethasone group. The 28-day mortality rate was 27.1% for those receiving 12 mg of dexamethasone, versus 32.3% for the 6 mg dexamethasone group. At 90 days, the mortality rate was 32.0% in the group given 12 mg of dexamethasone, against 37.7% in the group receiving 6 mg of dexamethasone. Major adverse effects, such as septic shock and invasive fungal infections, were reported in 11.3% of patients in the 12 mg dexamethasone group compared to 13.4% in the 6 mg dexamethasone group.	In patients with COVID-19 experiencing severe hypoxemia, administering 12 mg/day of dexamethasone as opposed to 6 mg/day did not significantly increase the number of days alive without life support within a 28-day period.
External corroboration that corticosteroids may be harmful to septic shock endotype A patients [33] 2021	Yes	97	Utilizing transcriptomic information from the VANISH trial, research subjects were categorized into pediatric septic shock endotypes A or B. This classification was based on the comparison of each subject’s gene expression profiles with the reference profiles for endotypes A and B [34]. The study aimed to explore the theory that hydrocortisone therapy correlates with higher mortality rates in patients classified under endotype A.	At the start of the study, there were no notable differences between the two endotypes. Patients with endotype A who received hydrocortisone had a mortality rate of 46%, in contrast to a 22% mortality rate for those with endotype A who were given a placebo. For patients with endotype B, the mortality rate was 22% for those treated with hydrocortisone and 19% for those receiving a placebo. The comparison between groups indicated that the likelihood of death was three times higher for endotype A patients treated with hydrocortisone compared to endotype B patients who received a placebo. No significant difference in mortality risk was observed between endotype B patients on placebo and either endotype A patients on placebo or endotype B patients on hydrocortisone.	This investigative analysis offers additional support for the possibility that exposure to corticosteroids could be linked to a higher mortality rate in patients with septic shock endotype A.
Effect of Ascorbic Acid, Corticosteroids, and Thiamine on Organ Injury in Septic Shock [35] 2020	YES	200	Participants were randomly allocated to either receive an intravenous combination of ascorbic acid (1500 mg), hydrocortisone (50 mg), and thiamine (100 mg) at 6 h intervals for four days or a placebo in equivalent volumes administered at the same intervals.	Over the 72 h following enrollment, no statistically significant difference was observed between the treatment and placebo groups regarding the change in SOFA score over time. There was no statistically significant difference in the occurrence of kidney failure between those receiving the intervention and those given a placebo, nor was there a difference in 30-day mortality rate. The most frequently reported serious adverse events included hyperglycemia (12 cases in the intervention group versus 7 in the placebo group), hypernatremia (11 in the intervention group versus 7 in the placebo group), and the onset of new hospital-acquired infections (13 in the intervention group versus 12 in the placebo group).	For patients experiencing septic shock, the use of a treatment regimen combining ascorbic acid, corticosteroids, and thiamine, as opposed to a placebo, did not lead to a statistically significant decrease in the SOFA score within the initial 72 h post enrollment. Based on these findings, the regular application of this combination therapy for septic shock patients is not endorsed.
Effect of Vitamin C, Hydrocortisone, and Thiamine vs. Hydrocortisone Alone on Time Alive and Free of Vasopressor Support Among Patients With Septic Shock: The VITAMINS Randomized Clinical Trial [36] 2020	YES	216	Participants were allocated randomly to either the treatment group, which received an intravenous combination of vitamin C (1.5 g every 6 h), hydrocortisone (50 mg every 6 h), and thiamine (200 mg every 12 h), or to the control group, which was administered only intravenous hydrocortisone (50 mg every 6 h). This regimen was continued until the resolution of shock or for a maximum of 10 days.	The duration of survival without the need for vasopressors by the 7th day was 122.1 h for the treatment group and 124.6 h for the control group. Among the 10 secondary outcomes predefined for examination, 9 exhibited no statistically meaningful differences. The mortality rate at 90 days stood at 28.6% (30 out of 105) in the group receiving the intervention and 24.5% (25 out of 102) in the control group. No incidents of serious adverse effects were documented.	For patients experiencing septic shock, administering a combination of intravenous vitamin C, hydrocortisone, and thiamine did not markedly extend the period they remained alive without the need for vasopressors over 7 days, in comparison to those treated with only intravenous hydrocortisone. This outcome indicates that the combined treatment regimen does not facilitate a quicker recovery from septic shock than the use of intravenous hydrocortisone by itself.
The cost-effectiveness of adjunctive corticosteroids for patients with septic shock [37]. 2020	YES	1513	The quality of life related to health 6 months post treatment was assessed using the EuroQoL 5-dimension 5-level questionnaire. Information on the utilization of hospital resources and associated expenses was gathered by integrating the ADRENAL dataset with governmental health administration databases. Measured clinical outcomes encompassed mortality, health-related quality of life, and the addition of quality-adjusted life years. Economic outcomes considered were the use of hospital resources, associated costs, and cost-effectiveness from the viewpoint of the healthcare payer.	No significant difference was observed in mortality rate or overall hospital expenses between the group receiving hydrocortisone and that given a placebo. The additional cost associated with hydrocortisone, for each quality-adjusted life-year gained, amounted to A $1,254,078. In the case of female patients, hydrocortisone proved to be cost-effective in 46.2% of bootstrapped replicates, while for male patients, it was deemed cost-effective in only 2.7% of these replicates.	The use of hydrocortisone as an additional treatment did not significantly impact long-term mortality, health-related quality of life, the utilization of healthcare resources, or costs, making it improbable to be considered cost-effective.
Hydrocortisone Compared with Placebo in Patients with Septic Shock Satisfying the Sepsis-3 Diagnostic Criteria and APROCCHSS Study Inclusion Criteria: A Post Hoc Analysis of the ADRENAL Trial [38] 2019	YES	2855	A subsequent analysis of the ADRENAL database was conducted to pinpoint patient groups that fulfilled the Sepsis-3 criteria for septic shock (termed ADRENAL–Sepsis-3) or the inclusion criteria for APROCCHSS (referred to as ADRENAL–APROCCHSS).	In patient groups from the ADRENAL study who matched the criteria for either Sepsis-3 or APROCCHSS, a greater mortality rate was observed at 90 days. However, hydrocortisone treatment did not significantly reduce mortality compared to placebo. The three groups showed similar results in several secondary outcomes, including a quicker reversal of shock with hydrocortisone, the frequency of mechanical ventilation resumption, the number of days spent alive and outside the hospital, and the occurrence of new bacteremia or fungemia. In both patient subsets receiving hydrocortisone, there was an increased incidence of shock recurrence, yet the impact on rates of blood transfusion did not differ by treatment. For patients who conformed to the Sepsis-3 criteria, those treated with hydrocortisone experienced a decrease in mortality at 28 days, an extension in the duration of being alive without the need for mechanical ventilation or renal replacement therapy, and more days spent alive and outside of the ICU.	Among participants in the ADRENAL trial who met the criteria for Sepsis-3 or APROCCHSS, administering hydrocortisone via continuous infusion did not significantly reduce the 90-day mortality rate compared to a placebo in cases of septic shock.
Transcriptomic Signatures in Sepsis and a Differential Response to Steroids. From the VANISH Randomized Trial [39] 2019	YES	176	A post hoc analysis was performed of a double-blind, randomized clinical trial in septic shock (VANISH [Vasopressin vs. Norepinephrine as Initial Therapy in Septic Shock]) [40]. Participants were enrolled within 6 h following the onset of shock. They were then randomly assigned to receive either norepinephrine or vasopressin, which was followed by the administration of hydrocortisone or a placebo. Comprehensive gene expression analysis was carried out across the genome, and the SRS endotype was identified through a pre-established model that utilizes seven distinct genes for discrimination.	No significant correlation was found between the SRS group and the type of vasopressor used; however, a significant interaction was observed between the allocation to hydrocortisone or placebo and the SRS endotype. Specifically, the use of hydrocortisone was linked to higher mortality rates in individuals exhibiting the SRS2 phenotype.	The gene expression pattern observed at the beginning of septic shock was related to how patients responded to corticosteroids. Individuals characterized by the immunocompetent SRS2 endotype experienced a notably higher mortality rate when treated with corticosteroids versus a placebo.

**Table 2 biomedicines-12-00984-t002:** Corticosteroid treatment in the selected studies—dose and duration of treatment.

Study/Year Published	Corticosteroid Dosage	Duration of Corticosteroid Treatment
Resuscitation With Vitamin C, Hydrocortisone, and Thiamin in Children With Septic Shock: A Multicenter Randomized Pilot Study [27] 2024	1 mg/kg per dose of hydrocortisone (with a maximum of 50 mg per dose) intravenously every 6 h immediately after patients were randomized in combination with vitamin C and thiamine.	The treatment was continued for 7 days or until any of the following occurred: the shock was resolved, the patient died, the patient was discharged from the pediatric intensive care unit (PICU), or if any major adverse events attributed to the treatment were observed.
Evaluation of hydrocortisone, vitamin C, and thiamine for the treatment of septic shock: a randomized controlled trial (The HYVITS trial) [28] 2023	50 mg of hydrocortisone every 6 h over 3 days, in combination with vitamin C and thyamine.	For 7 days or until ICU discharge, followed by gradual reduction.
Effects of hydrocortisone combined with vitamin C and vitamin B1 versus hydrocortisone alone on microcirculation in septic shock patients: A pilot study [29] 2023	Hydrocortisone (200 mg) was continuously pumped intravenously for 24 h in combination with vitamin C and thiamine or as a standalone treatment.	The infusion time was 30–60 min, and the interval was 6 h; 4 doses were used nt.
Early administration of hydrocortisone, vitamin C, and thiamine in adult patients with septic shock: a randomized controlled clinical trial [31]. 2022	Hydrocortisone (200 mg daily) along with vitamin C and thiamine vs. placebo.	For 5 days or until patients were discharged from the ICU.
Effect of 12 mg vs. 6 mg of Dexamethasone on the Number of Days Alive Without Life Support in Adults With COVID-19 and Severe Hypoxemia [32] 2021	Two study groups Received 6 mg of dexamethasone. Received 12 mg of dexamethasone.	
Effect of Ascorbic Acid, Corticosteroids, and Thiamine on Organ Injury in Septic Shock [35] 2020	Intravenous combination of ascorbic acid, hydrocortisone (50 mg), and thiamine at 6 h intervals vs. placebo.	4 days
Effect of Vitamin C, Hydrocortisone, and Thiamine vs. Hydrocortisone Alone on Time Alive and Free of Vasopressor Support Among Patients With Septic Shock: The VITAMINS Randomized Clinical Trial [36] 2020	Intravenous combination of vitamin C hydrocortisone (50 mg every 6 h), and thiamine vs. control group intravenous hydrocortisone (50 mg every 6 h).	This regimen continued until the resolution of shock or for a maximum of 10 days.

## Data Availability

Data sharing is not applicable.

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
