# Peer review of "Recent Data about the Use of Corticosteroids in Sepsis—Review of Recent Literature"

_biomedicines, 2024, doi:10.3390/biomedicines12050984_

Round 1
Reviewer 1 Report
Comments and Suggestions for Authors
This is a very interesting narrative review by Alexandra Lazar concerning the ambiguous effects of systemic corticosteroids in sepsis and septic shock. It is true that the use of corticosteroids as adjunctive treatment for sepsis has remained a very controversial topic. Although a systematic review and meta-analysis would be preferable the present paper summarizes rather adequately the recent data on the use of corticosteroids in sepsis and septic shock. However, I have two major comments:
1. I think that the inclusion criteria should be stated more precisely. For example, why was the following study by Marik et al ignored:
Marik PE, Khangoora V, Rivera R, Hooper MH, Catravas J. Hydrocortisone, Vitamin C, and Thiamine for the Treatment of Severe Sepsis and Septic Shock: A Retrospective Before-After Study. Chest. 2017 Jun;151(6):1229-1238. doi: 10.1016/j.chest.2016.11.036.
2. In my opinion the discussion section should be more extensive including a more critical view of the data. Arguments against the use of corticosteroids in sepsis are missing
Reviewer 2 Report
Comments and Suggestions for Authors
The utilization of glucocorticoids in sepsis is aimed at alleviating the excessive inflammation, potentially averting further tissue damage and enhancing outcomes. Nevertheless, the clinical evidence regarding their effectiveness is inconclusive, as some studies suggest benefits in specific patient subsets while others indicate no significant impact or even adverse effects.
The review by Lazar is somewhat unbalanced in the sense that 6/14 pages of text are devoted to tables with data from the literature. Maybe reconsider moving some stuff from the tables to the text.....
Round 2
Reviewer 1 Report
Comments and Suggestions for Authors
I think that the authors have adequately addressed my comments in the revised version of the manuscript. Therefore, I have no further comments.